# Cisnormativity as a structural barrier to STI testing for trans masculine, two-spirit, and non-binary people who are gay, bisexual, or have sex with men

Mackenzie Stewart[1], Heeho Ryu[1], Ezra Blaque[1,2], Abdi Hassan[1,2], Praney Anand[1], Oralia Gómez-Ramirez[3,4,5], Kinnon R. MacKinnon[6], Catherine Worthington[7], Mark Gilbert[3,4], Daniel Grace[1]*

1 Dalla Lana School of Public Health, University of Toronto, Toronto, Ontario, Canada, 2 Factor-Inwentash Faculty of Social Work, University of Toronto, Ontario, Canada, 3 BC Centre for Disease Control, Vancouver, British Columbia, Canada, 4 School of Population and Public Health, University of British Columbia, Vancouver, British Columbia, Canada, 5 Canadian HIV Trials Network, Vancouver, British Columbia, Canada, 6 School of Social Work, York University, Toronto, Ontario, Canada, 7 School of Public Health and Social Policy, University of Victoria, Victoria, British Columbia, Canada

* daniel.grace@utoronto.ca

**Data Availability Statement:** All relevant data are within the paper and its Supporting information files.

## Abstract

Trans masculine, two-spirit, and non-binary people who are gay, bisexual or otherwise have sex with men (TGBM) are under-tested for sexually transmitted infections (STI) and may face complex, intersectional barriers that prevent them from accessing STI testing. As part of a study on gay, bisexual and other men who have sex with men's (GBM) experiences of current STI testing systems in Ontario, Canada, this paper reports on the findings from TGBM participants' experiences with in-person STI testing in a range of venues (i.e. Family doctors, walk-in clinics, and community-based organizations) to explore testing barriers specific to TGBM. Using a community-based research approach, between June 2020 and December 2021 peer researchers who identified as GBM conducted focus groups and interviews with 38 cis and trans GBM, 13 of whom identified as TGBM. Data were analyzed following grounded theory. When questioned about past experiences with testing, TGBM participants reported several barriers to STI testing within current testing models in Ontario due to cisnormativity and heteronormativity. Cisnormativity is the assumption that everyone identifies as the gender they were assigned at birth, and those who do not are considered "abnormal", while heteronormativity is when it is assumed that everyone is heterosexual. From our research we identified three overarching themes concerning testing barriers among TGBM participants: (1) non-inclusive clinic environments, (2) lack of provider knowledge and competency, and (3) legal documentation. Inherent cis and heteronormativity in healthcare institutions appear to be factors shaping the historical under-testing for STI in the TGBM population. These findings suggest the relevance of implementing trans-specific clinical practices that reduce the stigma and barriers faced by TGBM in STI testing contexts, including: hosting all-gender testing hours, opening more LGBTQ+ clinics, offering training in transgender health to testing providers, and conducting a review of how gender markers

**Funding:** This work was supported by the Canadian Institute of Health Research (CIHR) [Implementation Science Team Grant: FR# CTW-155387; PIs: MG, DG, CW] https://cihr-irsc.gc.ca/e/193.html. DG is supported by a Canada Research Chair in Sexual and Gender Minority Health https://www.chairs-chaires.gc.ca/home-accueil-eng.aspx. AH was supported by the Richard B. Splane Social Policy and Social Innovation Scholarship https://socialwork.utoronto.ca/awards/richard-b-splane-social-policy-and-social-innovation-scholarship/ The funders had no role in study design, data collection and analysis, decision to publish, or preparation of the manuscript.

**Competing interests:** The authors have declared that no competing interests exist.

on health documents can be more inclusive of trans, two-spirit, and non-binary communities.

## Introduction

Trans masculine, two-spirit, and non-binary people who are gay, bisexual, or otherwise have sex with men (TGBM) encompass a wide variety of identities, body types, and sexual practices. The term "trans masculine and non-binary" is used to be non-prescriptive and indicate participants whose identified gender is different than their assigned sex at birth. In this paper, it refers to participants who were assigned female at birth who may identify as trans men, two-spirit or non-binary, and participants who were assigned male at birth and identified as two-spirit or non-binary. We also acknowledge the complicated and imprecise definition of the term "two-spirit" and that it is ahistorical and meant as a reclamation of Indigenous identity against the terms of colonization [1]. Two-spirit is intended to be an umbrella term for Indigenous peoples who construct their identities outside of colonial binaries, and we understand that its use extends beyond identification of both sexual and gender identity making it a complicated, cultural term that is used differently across various Indigenous tribes and communities [1]. Here, two-spirit people were included in the TGBM sample if they indicated that their identified gender was different than what they had been assigned at birth.

Despite these variations in terminology and identity, many TGBM have similar risk profiles for sexually transmitted infections (STI) as cisgender gay, bisexual, and other men who have sex with men (GBM) [2]. However, TGBM may require specialized care and different tests in comparison to cisgender GBM, depending on their body type and what kind of sex they are having. The US Centers for Disease Control recommends that sexually active GBM should screen for STI every three months to a year, depending on frequency and number of partners [3, 4]. Canadian national STI testing guidelines do not include specific guidance around risk factors or testing differences for transgender people or GBM [5]. Those who engage in penetrative vaginal sex require vaginal gonorrhea and chlamydia testing [4]. Additionally, TGBM who have cervixes should also be tested for HPV and other cervical cancers [2]. Notably, there is no differentiated guidance for non-binary and two-spirit people who have a penis and have sex with men from cisgender GBM. TGBM have a range of testing needs, and literature about which tests should be conducted for this population note that patients should be assessed on a case-by-case basis depending on genital status, hormone use, and sexual behavior history [2].

The limited available data currently indicates that TGBM are under-tested for STI while having a similar risk profile as their cisgender MSM counterparts [6, 7]. Despite having high levels of condomless sex, approximately 43% of TGBM in Ontario had never been tested for HIV, leading to possible underdiagnosis of HIV in the TGBM population in Ontario [8]. In the general Canadian GBM population, approximately 17% had never been tested, a dramatic difference from their TGBM peers in Ontario [9].

The trends of transgender men, including those who have sex with only women, being under-tested also have been tracked worldwide. In Thailand a sexual health center that services only transgender clients, only 5% of service users were trans masculine people who primarily accessed the clinic for gender-affirming hormones, not for STI testing [10]. Most studies that analyze transgender sexual health focus on trans women or people assigned male at birth [11]. This lack of data includes the CDC citing only trans women's risk, prevalence and prevention on their website and reports [12]. This dearth of data indicates that the TGBM community

currently lacks uptake of available services and needs targeted interventions to increase access to STI testing [11]. The impact of targeted programming would be to diagnose more STI in TGBM and provide accurate rates of STI in this community.

While TGBM have similar risk factors as cisgender GBM for STI, TGBM face population-specific barriers for STI testing as well as general barriers that affect GBM [13–15]. *General barriers* to testing for GBM include fear of the consequences of a positive test, social stigma, distrustful patient-provider relationships, lack of discretion in services, accessibility of health services, and perceptions of risk [15–18]. However, these barriers do not account for intersectional identities such as race, gender, class or ability that exist within the GBM community that create challenges to testing for those who exist in the margins due to their lived identities.

There is a paucity of data surrounding STI research for TGBM, including the barriers preventing this population from accessing testing [8, 19]. Often, studies on TGBM infection risk are focused on HIV, and the literature suggests an estimation of HIV prevalence in all trans men is between 0% to 3%, with 0.6% of Ontarian trans men reporting an HIV infection [20–22]. Previous studies in Ontario noted that 45% of trans men engaged in condomless anal or vaginal sex with a partner of unknown serostatus, which is considered a high-risk activity for contracting HIV [20, 23]. In 2018 the United States Centers for Disease Control (CDC) identified that .5% of 2,364 trans men in their sample were diagnosed with HIV, which is more than double the rate of cisgender women at .2% out of 4,753,672, and less than cisgender men at .9% out of 4,534,426, and far less than trans women who had the highest rate of HIV diagnosis at 2.7% of 13,154 [24]. We are using the 2018 data because there have been no updated statistics on the CDC website since 2018 about diagnosis rates in trans men. The 2018 statistics may be inaccurate due to the noted lack of testing in this population and the fact that in these categories in the number of trans men in their sample was exceedingly lower than the other populations being sampled. Additionally, these numbers are inclusive of HIV only and no other STI, which does not give a broad or accurate look at trans men's risk levels of STI acquisition.

The data related to TGBM and STI that are not HIV primarily come from convenience samples. Participants in a study in New England also noted that they considered themselves to be "moderately high risk" for acquiring an STI [25]. Other studies indicate that TGBM have similar risk levels of contracting STI as cisgender MSM and have engaged in high-risk activities such as condomless sex [26, 27]. While it is difficult to find current population-level surveys of STI diagnosis statistics of TGBM, one study of 45 TGBM that took place across the United States found that 46.7% of participants had been diagnosed with at least one STI that was not HIV in their lifetime, and 2.2% had been diagnosed with HIV [28]. A longitudinal study of transmasculine youth (12–29) in Boston found that 8.4% of trans men reported being diagnosed with an STI between 2001–2010 [29]. Outside of the United States, a Dutch study with a random sample of trans people assigned female at birth found approximately 6% of participants tested positive for chlamydia at the time of the study, although none tested positive for gonorrhea or HIV, indicating that more research needs to be conducted analyzing this population's risk for contracting chlamydia as it may be higher than other STI [30].

STI testing has unique barriers for TGBM in relation to testing environments and experiences because of *trans-specific barriers* to healthcare [31]. These barriers are a factor in the under testing of the TGBM community. Trans healthcare is rarely addressed in institutional settings or medical training, leading practitioners to believe that they must have specialization in trans care to provide any healthcare to trans patients, not just trans-specific care [32]. In one study, 83% of trans people with primary healthcare providers did not feel comfortable discussing trans issues with their doctors [33]. STI testing presents its own unique set of barriers for TGBM including: the fear of positive results delaying transition, mistrust of sexual health providers based on previous negative experiences, perceived low risk of trans patients

acquiring STI by providers due to the lack of knowledge of TGBM sex practices, assumption of clients being cisgender leading to lack of adequate care, and providers not having trans-specific sexual health knowledge which impedes testing or leads to lack of appropriate testing for TGBM [28, 34].

There is currently a lack of data that encompasses TGBM's experiences with STI testing, and the available data focuses heavily on rates of HIV testing and infection [34]. When addressing barriers to STI testing for trans populations, HIV testing alone cannot tell the entire story of barriers to all testing. Our data reports that HIV testing and infection rates, while valuable information, excludes important data related to the causes of discomfort and lack of other STI testing in this population. STI other than HIV require different testing mechanisms that affect TGBM's acceptability of being tested, including some that may induce dysphoria for some TGBM, leading to a lower level of acceptability for types of tests such as genital swabs. HIV tests can be performed using a blood draw or point-of-care test, giving TGBM the ability to not disclose their trans status to healthcare providers when being tested for HIV [35]. However, when seeking testing for STI such as chlamydia and gonorrhea, genital swabs are more effective than urine tests in diagnosis and as noted above, should be ordered for patients if going through routine testing [36].

We were unable to identify any current research that indicated the difference in acceptability between STI tests with genital swabs, and those that do not require genital swabs for TGBM. However, it is the anecdotal experience of the authors who identify as TGBM, as well as an inductive finding through our research that these differences in testing methods may affect testing frequency and increase testing avoidance amongst TGBM. Additionally, previous studies examining trans people's experiences with reproductive healthcare indicated that internal examinations or insertions may present a barrier for trans people accessing health resources due to their perceived incongruence with a patient's gender identity. One study found that birth control methods that require insertion, such as an IUDs or vaginal rings, may cause or aggravate dysphoria for some trans people [37]. Another study examining the experiences of trans people with cervixes accessing pap tests reported that a major barrier to trans participants who sought testing was reconciling their masculine identity and what they saw as a 'feminizing' or 'feminine' procedure, and those who were on the far end of masculinity struggled with the gender dysphoria they faced from a pap test [38]. Based on these studies, we conclude that similar barriers may be faced due to internal STI swabs in the TGBM community and believe there should be further research and action to reduce these the distress internal exams may cause for TGBM patients.

Our objective was to examine TGBM's experiences of STI testing services at their family doctors, walk-in clinics, community-based organizations, or other in-person testing available in Ontario, Canada at the time of the study. In analyzing TGBM's STI testing experiences, we discovered multiple barriers that may impact testing rates in this community and will provide recommendations to relieve these barriers.

## Methods

Our TGBM data are part of a larger study examining cis and trans GBM's STI testing experiences in Ontario. The objective of the larger study was to gauge the acceptability and population-specific benefits of an online testing program. We used the online testing program GetCheckedOnline (GCO), currently available in parts of British Columbia, Canada, as a model for what online testing could look like in Ontario. GCO is currently operated by the British Columbia Centre for Disease Control (BCCDC). The process of testing with GCO is as follows: those seeking testing create an account on getcheckedonline.com; fill out a sexual

behavior-based questionnaire; have tests recommended to them based on their behavior; they are sent an electronic lab requisition that they can then bring into select locations of Lifelabs, a Canadian-based lab corporation; they present the lab requisition on their phone or by paper, they have their specimens collected and are notified through the online system if their results are negative. If results are positive or inconclusive, they are given the number of a public health nurse to contact who will connect them with further care and resources.

The first phase of focus groups and interviews were conducted between June-September 2020, and a second phase conducted between September-December, 2021.The second round comprised of Black and Indigenous participants only, specifically to capture the unique experiences of these communities that were originally underrepresented in the first round of data recruitment and data collection. This study was conducted following community-based participatory research (CBPR) principles, a method that values partnerships from organizations and members of the communities being studied during the research process to promote the well-being of these communities [39]. As part of the CBPR process, four peer researchers who identified as part of the GBM community were hired to conduct focus groups and synthesize data. To anonymize the data, each participant was assigned a pseudonym which was used throughout data analysis.

We worked closely with an Ontario-wide community advisory board (CAB) of GBM that included public health workers, therapists, AIDS Service Organization (ASO) workers, and GBM who had experience with or an interest in testing innovations. The CAB was consulted during data collection, analysis, and interpretation phases. Ethics approval for this study was received from the University of Toronto Research Ethics Board.

## Recruitment

Participants were recruited through purposive sampling, a method commonly used to recruit marginalized and hard to reach communities using targeted networks [40]. Purposive sampling was used to recruit GBM to ensure a range of diverse experiences were represented. Flyers advertising the study were posted online on Facebook, Instagram, and Twitter to attract potential participants. Recruitment posters were also distributed through local Pride Groups, ASOs, and other community-based organizations (CBOs) that serve sexual and gender minorities within Ontario. Potential participants were invited to complete an eligibility screening questionnaire on SurveyMonkey, and those who qualified were invited to participate.

When advertising for the study, terminology on recruitment materials asked for "cis and trans gay, bisexual, queer and other men who have sex with men". Eligibility criteria stated that participants must be >18, identify as GBM, and tested for STI in the 12 months prior to the study.

## Data collection

Due to social distancing public health measures introduced during the COVID-19 pandemic, data collection took place online using Microsoft Teams and Zoom. A $30 CAD honorarium was provided to each participant. Peer researchers conducted 60–90-minute semi-structured focus groups and interviews with cis and trans GBM who had tested for STI within the past twelve months in Ontario, Canada (n = 38) to understand their experiences of current testing models in the province. We conducted 10 focus groups (2–6 participants each) and 7 individual interviews.

Prior to the interviews, participants provided their written, informed consent and completed a sociodemographic survey on Qualtrics. During the session, participants were shown a video describing GCO in British Columbia and were asked a number of questions surrounding

the acceptability of an online testing model, as well as to describe previous testing experiences they had with traditional in-clinic testing [41]. These questions queried positive testing interactions, negative testing interactions, barriers to being tested in these spaces, and participants' ideal testing environments.

## Data analysis

Audio recordings of focus groups and interviews were transcribed verbatim. Participants were assigned a pseudonym for anonymity. Each session was conducted with at least two Peer Researchers present to moderate the focus groups and take notes. Peer Researchers then coded the transcripts using NVivo 12 software by creating a universal codebook and coded transcripts separately based on these codebooks. We conducted our analysis using a grounded theory approach [42].

Analysis and writing were led by a peer researcher who identifies within the TGBM community, with support from a diverse team of cis, trans, and non-binary GBM researchers. After developing the initial codes, transcripts were sorted into cis and trans GBM for the purposes of this analysis based on emerging findings from initial coding. The TGBM subset of transcripts were subsequently collated to create TGBM-specific themes to explore whether there was a cohesive dataset that spoke to commonalities within this population given their unique set of experiences. Finally, the entire research team gathered to review these themes and revise codes relevant to this manuscript, led by the senior author.

## Results

### Participant characteristics

Of the 38 participants interviewed in the study, 13 participants (34%) identified as either trans, non-binary or two-spirit, and are the focus of this analysis. Most trans participants (62%, n = 8) identified as trans men. Two participants selected 'prefer not to say' in their demographic screener, but discussed the ways testing experiences were impacted by their trans identities in the focus group and interview they participated in. Forty-two percent of participants were white (n = 6), and participants ages ranged between 24–45 years old. There was also a diversity of GBM identities captured in the sample, such as gay, bisexual, pansexual, queer and asexual. The majority of participants had a bachelor's degree or higher (62%, n = 8). Almost all participants had been tested for STI other than HIV in 12 months prior to their participation (84%, n = 11), and almost all had been tested for HIV in the 12 months prior to the study (92%, n = 12). Comprehensive demographic information is included in Table 1.

While some participants noted positive experiences with STI testing, the majority faced barriers to testing related to their TGBM identity. Our results revealed that cisnormativity and heteronormativity played an integral role in the barriers TGBM participants faced when accessing STI testing institutionally and with individual providers. Cisnormativity is expectation that all bodies and people align with the gender they were assigned at birth and is an underlying process of erasure for trans people. Cisnormativity implies that cisgender people are the "default" bodies in our society and labels bodies such as intersex or trans people as 'deviant' or 'abnormal' [43]. In a testing environment cisnormativity can be seen in the assumption that all men have penises and all women have vaginas, or that all clients' sex is congruent with their legal gender on their health cards, lab forms, or any other documentation presented to the provider [43]. Heteronormativity, assuming that everyone is heterosexual, also played a part in provider's inadequate treatment of participants by assuming relationship configurations or that patients only have sex with people of a different gender [43]. Taken

together, cis- and heteronormativity is the view gender is related to biological sex, and that sexual attraction may only be to those of an opposite sex [43].

We identified three common interrelated barriers to STI testing based on participating TGBM's accounts: *cisnormative testing environments*, *lack of provider competency in trans*

**Table 1.  TGBM participant demographics** *(n = 13)*.

|  | n | % |
|---|---|---|
| **Gender**[*] | | |
| Trans | 8 | 62 |
| Non-Binary | 4 | 31 |
| Two-Spirit | 2 | 15 |
| Prefer not to say | 2 | 15 |
| **Age** | | |
| 18–24 | 3 | 23 |
| 25–34 | 6 | 46 |
| 35–44 | 3 | 23 |
| 45–54 | 1 | 8 |
| **Race/Ethnicity** | | |
| white | 6 | 46 |
| Indigenous/First Nations | 3 | 23 |
| Black | 1 | 8 |
| Black/white | 1 | 8 |
| South East Asian | 1 | 8 |
| Latin American | 1 | 8 |
| **Sexuality**[*] | | |
| Bi/Pan | 6 | 46 |
| Gay | 5 | 38 |
| Queer | 3 | 23 |
| Asexual | 1 | 8 |
| Straight | 1 | 8 |
| **Highest Level of Education:** | | |
| Did Not Complete High School | 1 | 8 |
| High School or Equivalent | 3 | 23 |
| Post Secondary (diploma/certificate) | 3 | 23 |
| Bachelor's Degree | 4 | 31 |
| Above Bachelor's Degree | 4 | 31 |
| **Last Tested for STIs Other than HIV** | | |
| Within 3 months | 7 | 54 |
| 4–6 months | 1 | 8 |
| 7–12 months | 1 | 8 |
| 12+ months | 1 | 8 |
| Never | 1 | 8 |
| **Last Tested for HIV** | | |
| Within 3 months | 5 | 38 |
| 4–6 months | 5 | 38 |
| 7–12 months | 2 | 15 |
| 12+ months | 1 | 8 |

[*] Percentages may not total 100; categories not mutually exclusive.

*healthcare*, and *documentation barriers*. These barriers were specific to TGBM, and were compounded by the barriers we found in both cis and trans GBM, which included: inaccessible or inconvenient clinic hours, lack of anonymity in testing environment, fear of stigma from providers about sexual behaviors or practices, as well as low levels of trust in providers to be informed about GBM identities. All barriers were identified based on participants' past experiences with in-clinic testing, which included testing with family physicians, public health testing sites, sexual health clinics, walk-in clinics, and ASOs or other CBOs.

## Cisnormative testing environments

**Gender segregated clinics.** Participants described STBBI testing environments as exclusionary of trans and gender non-conforming people within their procedural policies. *Gender segregated clinic hours* was named most often as an exclusionary policy that relied on cisnormative expectations that clients identified with one gender, or that clients would automatically know which hours to attend if they were trans. Participants reported that some clinics sorted their testing into "men's" and "women's" hours, or "men and trans" or "women and trans" hours, which was misleading and unclear for trans, non-binary and gender non-conforming service users. Participants who wished to access these resources were confused about the wording of these hours, and often wondered if they fell into the "and trans" category or in the "men" category based on their gender identities, and if spaces were sorting them according to which genitals they had. Participants felt as though this ambiguity created uncertainty around the hours that TGBM should attend the clinic for and wondered if the services offered would fit their health and testing needs while being respectful of their genders.

Felix (24, white, trans male) described his experience with gender segregated clinics and the barriers he noted as a result of this model:

[The clinic I went to] have what they call a 'women and trans clinic' and 'men and trans clinic' and that always has a big question mark of 'oh, what services are where' and they don't clarify. But exactly sometimes it's like they might tell me as a trans man, 'Oh, but that service you need is [the] women and trans clinic'.

Ian (29, white, trans and non-binary) shared similar struggles as Felix when interacting with clinics that had gender-segregated hours:

And then trans-inclusive clinics—supposedly trans-inclusive clinics—LGBTQ centered clinics I guess, well I found that usually they had gendered hours. And even though it's trans-inclusive, I never actually knew which clinic to go to. I had a doctor during the men's hours assuming that I was a cis guy, and then I felt like weird and intrusive when I went to the women's hour. So I would sort of just not go even though that clinic was a lot closer to where I was going to school.

Felix also spoke to the fear of violence and transphobia in spaces that use gender-segregated models when he said, "And [a gender segregated clinic] already is an unsafe space. Just like being present there, you're kind of outing yourself". Responding to Felix in their focus group, Frederic (30, white, non-binary) agreed that the physical space of a waiting room could be a source of tension, as well as noted these gendered barriers could encourage fear from trans people who may be scared to sit in a room full of cisgender men.

**Trans-inclusive terminology.** A related barrier that was raised related to the use of gender-specific or sexuality-specific terms when naming testing hours. *Inclusive terminology* of TGBM identities that pushed back against cis and heteronormatvity was mentioned by

participants as important to their comfort when attending testing. Participants believed that inaccurate or ambiguous terminology such as "men" and "gay" for testing hours prioritized cisgender GBM and created an environment that may exclude TGBM from attending these testing hours:

> Like, I'm thinking of [a CBO] in particular, they focus on incorporating like non-cis guys and trans guys into their kind of mandate, and bi guys or there's lots of trans folks that like come to [name of a program targeted for gay men] who, maybe, wouldn't call themselves gay, or, you know, who knows, right? (Isaac, 24, Black, non-binary)

**Institutional cis and heteronormativity.** Another barrier related to clinic environments was the *structural cisnormativity and heteronormativity* in spaces such as walk-in clinics and clinics not specifically dedicated to LGBTQ+ populations that impeded participants from seeking testing. Often, the onus on the trans client to request proper tests and disclose their trans status because providers assumed their genders based on their appearance or inaccurate documentation. For example, Ian noted his experience with what he termed "normative" clinics, and while he specifically names heteronormatvity in these interactions; his experiences also reflect cisnormativity:

> So I was sometimes reluctant to go to the doctor and I would just sort of postpone. And previous to that, or during that period I also–occasionally would go to walk-in clinics, but I found them really difficult at like normative clinics. Like heteronormative clinics, I usually would find it intimidating to talk to the doctor and have to out myself as a trans person every time.

Ian's experiences of cis and heternormativity by providers created a barrier for him to feel as though he could not enter a clinical environment to be tested, because of the inherent danger that came with outing himself to an unfamiliar practitioner who might not be knowledgeable about Ian's health needs due to the intersection between his gender and sexual identity.

## Lack of provider competency in trans healthcare

Participants also noted that there was a lack of consistent trans-informed care from providers when accessing STI testing. Drawing on their lived experiences of accessing testing, participants explained that some service providers lacked the competency to know the sexual risk profiles of TGBM and did not provide comprehensive and informed care when testing them for STI.

**Provider education.** One common theme from multiple participants was *educating providers and performing self-advocacy*. TGBM participants felt as though their providers did not have enough education or training in trans care to provide them adequate care. Therefore, when communicating their testing needs and how testing was impacted by their transition status or risk level, participants had to be their own advocates to these providers. Sean (24, Black/white, trans/non-binary) stated that they often had to explain "how toys or whatever" work and the risk levels of these forms of sex practices.

Despite having to educate their own providers, participants lowered their standards of care and considered this education to minimally impact their relationship with their provider. Participants believed that if the provider was willing to listen to the education provided, they would continue to see their current providers. This is demonstrated when Isaac (24, Black, non-binary) chose a provider who might not be the best fit for their care, but were not stigmatizing their gender modality, a term described by Florence Ashley to capture the lived

experience of trans people's genders as they relate to the gender they were assigned at birth, after previous negative encounters with doctors [44]:

> [My] physician already knows that I'm queer and trans. I mean she doesn't always like 'get it' necessarily, but it's not a surprise, you know. And it's not as annoying to have to explain in terms of my risk profile, which it has been a problem with walk-in doctors or like other physicians I've had in the past.

Isaac discussed being willing to lower their expectations to accept care from a provider who "doesn't always get it" but was more accepting of their trans status than other providers they had seen previously. While this physician may not have been the best fit for Isaac's sexual health needs, and that there may be more trans-specific education needed in institutional spaces outside of a patient-provider relationship; the provider was seen as being an acceptable practitioner due to the *willingness to be educated* about Isaac's transness and accept the labor Isaac performed to educate the provider about their healthcare needs. Jaden (28, First Nation, trans/two-spirit) faced similar barriers when having to educate his doctor about the intricacies of trans bodies and health needs, "And something that made it hard to get tested is just body dysphoria and my doctor not knowing what happens to trans folks' bodies after hormones and things like that. So having to really educate in that moment to my doctor is a little bit awkward and uncomfortable".

George (39, white, trans male) relayed a demoralizing experience at a general health clinic with a provider who was not knowledgeable about how STI impacted TGBM's anatomy where he had to educate his provider and noted in his focus group that he felt as though he had to "steel himself" to see a healthcare provider due to negative previous encounters,

> "And like I remember one time I was at the public health clinic and I was trying to explain to them like, "No, Chlamydia can be [in a front] hole also. I don't know why I'm telling you this but just so you know".'

These accounts of practitioners not knowing proper tests, or ways that STI specifically affect TGBM speak to individual ignorance, but also to larger structural systems of cisnormativity. Listening to multiple accounts of providers lacking the knowledge to treat this population, it becomes clear that there was no formalized education structures for the providers to be taught about TGBM risks and population.

**Assumptions from providers.** Participants also spoke to *assumptive providers*, who made them feel discomfort with their doctors due to cis and heternormative assumptions. Instead of taking a patient-centered approach to testing, providers assumed that they were the experts in their patient's care and often incorrectly assumed important factors when testing for STI. Participants noted that this led to having to advocate for themselves in the face of what they believed to be under-testing or mis-testing. Isaac relayed their experiences with providers,

> In the past the homophobia and transphobia from different types of practitioners who like maybe don't understand, or who are confused about kind of like my risk profile for acquiring different STIs. So sometimes having to like fight these practitioners to be like 'I want a full panel of STI testing not just for, you know, like syphilis or gonorrhea'. Or, like you know, I guess them assuming-making assumptions about the type of sex I'm having and maybe suggesting that I'm maybe over exaggerating my risk.

Sean shared similar sentiments to Ian in their own interview, and stated that they had also felt as though it was personally difficult to be tested because their doctor underestimated their

risk for STI acquisition as a person with a vulva having sex with other people with vulvas, demonstrating cis-heteronormative assumptions of patients' risk levels by not questioning potential risks because of the anatomy of the patient and their partner.

Isaac also spoke about previous negative experiences with testing providers making them feel "uncomfortable" requesting testing due to homophobia and transphobia from providers, and how these providers stigmatizing their TGBM identity made testing more difficult.

**Lack of TGBM knowledge.** The final provider-related barrier we identified was *ignorance of TGBM's sexual health needs*. Many participants noted that providers who did not work in spaces that specialized in LGBTQ+ care did not have adequate knowledge of STI that TGBM may be more susceptible to acquiring.

Ivan (45, white, trans male) also noted that his experience with doctors who did not specialize in GBM health meant that the burden of information and knowledge was placed on him:

I asked a doctor who gave me results for something and they had no idea about the impact [of an] STI issue that's very specific to our community as transmen. And had it been the doctor or the specialist who was a gay man that works at the HIV clinic he would've been 'oh yeah, like you know, give [the test]', right.

Ivan expanded upon these barriers when he said, "So I think for me, the issues I have faced, it's just people's ignorance around what's involved in gay sex, particularly bottoming and trans people–different things that make it easier, more susceptible to certain STIs, things like that. So I think that's a big problem".

Felix (24, white, trans male) noted that he needed a sensitive provider who had knowledge of trans bodies and issues to feel comfortable with certain types of testing:

Definitely in terms of different types of testing like I have gotten an HIV rapid test because it's very like comfortable in how quick and easy it is but personally as a trans person I wouldn't go to just any doctor for every STI test. I feel like they would need to be informed. [. . .] I need to feel comfortable with the whole situation and I can't just be thrown into it.

For Felix, his trans status directly correlated to his level of trust toward providers. He felt not all providers would have the competency to provide adequate, informed trans care for more invasive genital or rectal swabs. Felix also noted the importance of consistent, trans-informed care for TGBM who may be hesitant to be tested due to anxiety about the level of care and knowledge from a service provider, "If you go to [name of a sexual health clinic] you just never know who you're gonna–like what person you're gonna see.[. . .] So you still have to go somewhere but for me, I–I'm scared of doctors". Here, Felix implies that while most providers would be able to technically administer the pharyngeal, vaginal, and rectal swabs as well as blood tests, having a consistent provider that he knows is trans-informed and capable of providing care would be preferred.

Participants also mentioned positive experiences with healthcare providers. The perceived acceptability of LGBTQ-specific healthcare clinics and service providers was common among TGBM participants, addressing the noted barriers from non-LGBTQ clinics and providers. Darren (27, Latin American, trans male) and fellow focus group participant Damian (39, white, trans male) discussed their positive experiences with providers:

Darren: Well the only thing that I want to share are my tests are not like forced because I can choose to have it, but they are sent in all the time from my doctor [it's] part of the–her service, like she's always taking care of them. [. . .]

Damian: Yeah, I can jump in. I think sort of similar to what Darren said that my doctor's always like "You need to get tested." It's not like they're forcing it, but they're always reminding me, and I think that comes from having a very like LGBT [competent] family physician.

In sum, many TGBM participants reported negative experiences with service providers due to the lack of knowledge surrounding their sexual health, and some participants expressed reluctance to engage with service provider-led STI testing. Those who felt positively about testing experiences reported having received care in clinics that offer specialized services to the LGBTQ/GBM population.

## Documentation barriers

Participants noted that when documentation, such as medical charts, lab forms, and health cards were incongruent with their gender, this created barriers to testing. TGBM who have documentation that do not reflect their gender are often misgendered in testing settings because of cisnormative ideas that everyone identifies with the gender and name on their legal documentation, or that all service users identify with their sex assigned at birth. In both these scenarios the burden of coming out as trans is placed on the client to receive healthcare in which the client's gender is affirmed and they receive correct tests based on their bodies and behaviors.

Isaac (24, Black, non-binary) did not have documentation that reflected their identified gender but felt comfortable using that documentation for their healthcare needs. However, they did note that other trans people may face barriers related to their identification:

I mean I am a trans person and my gender marker on my ID doesn't, you know it, it says female and that's like not necessarily how I would identify. And so, I can see how for other trans folks, that like if their gender marker doesn't match their actual identity how that might create issues or cause problems.

In a focus group, Sean (24, Black/white, trans/non-binary) spoke to their lived experience having a gender on their ID that is incongruent with their presentation. They characterized these experiences as "awkward" and had providers attempting to "figure out" which gender and sex they were. They also noted that there were little options to correct their name and gender in lab systems, leading to confusion from staff,

And the only trouble that I ever have in that experience is before my name was legally changed, there's no way to actually, I don't know if there's a way to have them call out the name I go by instead of your name on file. And the other problem that I have is that I'm always just like pointing to the part on the form that says F for female and asking if this is me.

Ian (29, white, trans/non-binary) echoed the documentation barriers Isaac and Sean identified. He had changed the gender on his health card to reflect his gender at the time of his interview, but when he had documentation that was incongruent his gender, he was less likely to access health services or testing:

Well definitely before I had my gender marker changed, that was pretty stressful to use my health card. And I tried to avoid [testing] as much as–as I could. I mean, or before I had a

name change it was also very stressful to use my health card. Those were kind of the major concerns I had personally.

Not wanting to show identifying information or documents due to safety concerns was also perceived as a potential barrier for trans people. Ian spoke to this, and connected his experience as a trans sex worker to reasons why privacy was important to him:

Yeah, I think like, um, as I stated before, I think that like there have been times in my life where that kind of privacy would be of concern for me—like if I was doing sex work again, or while I was doing that in my life, or at a time when I felt, um, like a little more distrusting of healthcare officials due to my trans identity.

Having gender on health identification and other documentation also led to assumptions of types of anatomy and sexual activity that service users have by providers. For trans people, having gender markers that are binary could not only be inaccurate to their identity and lived experience, but also cause false assumptions about tests needed, forcing the burden of disclosing their transness and advocating for proper tests on the service user. Ivan (45, white, trans male) explained,

. . .that makes it difficult, also the fact that . . . yeah, male on my ID, but ultimately I have to say–when I go for testing–that, you know, I am still biologically female, because there are certain things that I can face, you know, that they need to test for, HPV, things like that, cervical cancers, that if I said I was, you know–my health card clearly says male and I didn't let them know what was on it but I still had ovaries, etc. You know, particularly a lab, not your family doctor's office, then, you know, you're not getting tested for things like cervical cancer, sadly, that are serious.

Incongruent documentation created barriers for TGBM that forced the burden of coming out as trans onto the client, as well as perpetuated cisnormativity in testing environments for providers who may not have a client's full history and recommend tests solely on a client's presentation and gender marker.

## Discussion

TGBM face unique structural and individual challenges when accessing STI testing compared to cisgender GBM including: non-inclusive clinic environments such as gender-segregated spaces, lack of provider competency due to cisnormativity or lack of trans-specific knowledge, and documentation barriers for trans people whose legal documents do not reflect their presentation or assumed gender. While the majority of our TGBM participants discussed negative experiences with services for STI testing, those with positive experiences had been treated by providers with specific training in LGBTQ+ health issues and needs.

Participants identified multiple experiences with in-clinic testing where they experienced difficulties in access due to their identified gender, as well as the cis and heteronormative expectations in these testing environments. One barrier is the lack of sufficient information for where TGBM should seek testing in spaces that enforce gender-segregated hours. TGBM may attend the "men's" hours in accordance with their gender modality or physical appearance, but then be referred to "women's" hours to access testing that aligns with their anatomy and or assigned sex at birth. These referrals prolong the wait for testing at best; at worst the service user may be triggered by a referral to the clinic that does not align with their identity, or they may not have the time to seek a secondary testing appointment, causing them to not get tested

at all. Secondly, gendered clinic hours put trans people at risk of violence through forcing them to attend clinic hours of a gender they don't identify with. These clinics also exclude non-binary service users who may not want to claim the identity of 'man' or 'woman' and do not feel like they are represented in the term 'and trans', leaving them out of testing spaces entirely.

Service providers who could not provide trans and non-binary competent care were also perceived as a barrier to testing for TGBM. Paine coined the term 'embodied disruption' to describe the experiences of transgender men, gender non-conforming cisgender women and non-binary patients which reflected that healthcare providers often 'mis/recognize' patient's gender [45]. For trans men, this 'mis/recognition' often manifested itself as patients being assumed as a cisgender male, but they had to out themselves as trans to their service providers to receive proper affirming care.

We found that this concept of misrecognition was rooted in cisnormativity, which led to the assumption of clients' gender modality to be cisgender, or provider's absence of knowledge in trans care and created a lack of quality care for TGBM in testing environments. This lack of quality care caused some participants to encounter barriers including distress, or being forced to self-advocate when being tested for STI. Our participants related that they had to "settle" for healthcare providers who may not understand their gender modality and required their clients to educate and inform them on the sexual health and other risks for TGBM. This finding is consistent with noted limitations in medical and health professions education on caring for trans people [46, 47]. Beyond their own personal experiences of embodied disruption and being mis/recognized in an individual setting, these experiences pointed to broader systemic issues with healthcare settings that reflect the institutional erasure of trans healthcare needs in clinical settings that has already been well-documented for trans men, trans women, and non-binary individuals [32, 48, 49]. However, not all healthcare providers were perceived as lacking trans-specific knowledge. Participants believed that LGBTQ+ providers and spaces were more acceptable to be tested at and fostered more positive interactions with healthcare systems that addressed their specific needs as sexual and gender minorities.

Incongruent documentation was also perceived as a barrier for TGBM's access to STI testing. This incongruence led to missed testing for some participants, and inaccurate risk assessments of sexual behaviors for others. For those whose gender modality did not reflect their legal gender, seeking testing could potentially be triggering, feel difficult, or like an invasion of privacy. The feelings participants shared around incongruent health documentation are also of note due to the fact Ontario has not assigned gender on the Ontario Health Insurance Plan (OHIP) cards since June 13, 2016 [50]. This fact indicates that despite most participants likely possessing OHIP cards without gender markers displayed, identity documentation barriers were one of the primary noted experiences they felt they may encounter when accessing STI testing. However, these barriers can be explained upon further examination of the Ontario healthcare system's internal assignment of gender.

Health cards are what could be considered public-facing healthcare documents, which are held by the patient. These external, public-facing documents do not show a gender, but that is simply for the benefit of the holder. The Ontario Women's Justice Network (OWJN) noted that even upon the removal of gender on a health card, "the government continues to keep information about people even after they change names or genders. This means that your old gender marker and name may be kept in internal government records" [51]. While there are multiple resources for trans people in Ontario to change various forms of identification, such as METRAC, Rainbow Health Ontario, and the Ontario government website, none of these guides provide information about these internal markers within institutional systems [50, 52, 53].

The dissonance between public-facing documents having no gender, while internal systems assigning gender to all patients is notable for our study as it informs participant's perceptions of attending testing environments due to gender incongruence on health documentation. Places to get tested such as lab sites, doctor's offices, and sexual health clinics still assign clients a gender, male or female, within their internal systems that may not be representative of TGBM's lived experiences. This internal assignment creates barriers for TGBM as they must make the decision to either disclose their trans status to providers or choose not to disclose and risk being misgendered based on what providers see on these internal documents.

The findings in this paper confirm the barriers to HIV testing that had been previously identified for the TGBM community such as cisnormative practitioners assuming client genders, and lack of knowledge of TGBM's sexual health needs [28, 32, 54]. To our knowledge, there has only been one previous study that was inclusive of TGBM's experience of HIV testing and other STI such as syphilis, chlamydia, and gonorrhea [34]. This previous study identified barriers including lack of trans health knowledge among testing providers, limited clinical capacity to meet STI testing needs, and a perceived gap between trans-inclusive policies and their implementation in practice [34].

Our results dovetail findings from other studies related to TGBM's health. Previous studies have indicated that providers lack education on trans patients, not exclusive to TGBM, across healthcare [55–58]. As found in our study, having uneducated providers made trans people less likely to want to access care [58]. One study found that 24% of their sample of trans patients had to educate their healthcare providers about trans people and health over the course of their visit [58]. However, the other finding from this study indicated that this need to educate directly impacted trans patient's mental health [58]. Patients who had to educate their providers were more likely to experience depression and anxiety [58]. These findings are not exclusive to North America and have global implications; the World Health Organization conducted an assessment of transgender people in Asia and the Pacific and found that trans people who had access to healthcare providers were stigmatized and faced discrimination from their providers due to lack of trans knowledge [59]. Another study in Vietnam found that providers had knowledge gaps and were unprepared to treat transgender clients [60]. In general, healthcare for trans people is falling short of adequate and informed care, and sexual and reproductive healthcare follows this trend.

Pulice-Farrow et al. examined the barriers transmasculine patients of all sexual orientations face when accessing gynecological care and have results that mirror those in this study [61]. Specifically, they found transmasculine participants faced barriers related to providers' lack of knowledge surrounding transmasculine sexual health, or assumptions of risk levels and bodies that treated transmasculine people as a monolith without giving them individualized care. Our findings as well as other literature demonstrate that for TGBM, living at the intersection of being both a gender and sexual minority has created unique barriers for those who seek care from any provider, including sexual and reproductive health providers.

## Recommendations to facilitate uptake of STI testing in TGBM

Based on our findings, changes need to be implemented in places that conduct STI testing both in Ontario and to the broader community of providers who work with GBM to remove the identified barriers for TGBM and increase testing prevalence in this population. We identify three recommendations to facilitate better testing experiences for TGBM, that could also benefit the trans community at large.

The first recommendation is for clinics to de-gender some clinic hours. In practice, this recommendation would ask that clinics have general hours for all genders, which would allow for

non-binary people to also feel welcomed into the space without having to align themselves with binary genders such as 'men' and 'women'. This recommendation would also allow for trans people who do identify within a binary gender to receive treatment without having to navigate the barriers of aligning themselves with their gender modality or their assigned sex at birth. However, we recognize the benefits of having gender-specific hours for cis and trans women to attend women's only hours; therefore, we do not advocate for the complete removal of gender-segregated hours but recommend adding inclusive all-gender clinic times [62].

We also recommend more training in transgender health and knowledge for testing providers, as well as to establish more clinics dedicated to caring for the LGBTQ+ community. Multiple participants noted that the negative experiences they had faced with providers who did not have knowledge of trans health, or sexual health as it would pertain to a TGBM, were from general practitioners or walk-in clinics. Those who did speak about positive experiences in testing spaces generally noted that these experiences primarily took place in a space dedicated to sexual and gender minority health. However, this suggestion is for short-term changes to increase TGBM's comfort in testing and remove immediate barriers that TGBM face when seeking testing. A larger-scale and longer-term intervention would be increased training in transgender health for all healthcare providers to reduce cis and heteronormative assumptions in testing spaces. Increased training has been proven to effectively aid in the reduction of transgender stigma and transphobia in providers [47, 63]. Our recommendations, which are derived from our participant data, are congruent with other recommendations found in a community consultation with transgender people in 2015, in which the suggestions were given to 1) support a multi-disciplinary clinic for LGBT individuals; 2) create a network of LGBT-friendly service providers; and 3) give more training to providers and support staff [64].

Our final recommendation is to undergo a review for best practices to change gender markers within the publicly funded health system to create gendered data standards, and remove healthcare barriers for TGBM. While gender has been absent from public-facing OHIP cards for several years, service users are still gendered within internal systems that may not reflect their lived experiences or genders leading them to be misgendered and become more hesitant to access testing services and other healthcare. We acknowledge that gender markers on these documents are static texts that do not reflect the fluidity of identity, and cannot accurately reflect the gender modality of people holding these documents; more can be done to support trans people in reflecting their ideal gender representation on these documents. However, reconciling the needs of trans people and needs of public health surveillance need to both be accounted for when making any changes to the current systems.

Surveillance in public health, such as accumulating gendered data, is seen as necessary for programming for TGBM and other trans identities; as well as to address the institutional erasure of trans identities that can happen within these systems without the information demographic data provides. However, the use of these data can be troubled through the histories of what trans scholar and activist Dean Spade has termed "administrative violence", which details how administrative documents work to determine who is allowed to move through public spaces, and has created a lack of trust in governmental systems who ask trans people to identify their gender to the government [65]. Particularly, these forms of identification and gender become more complicated for racialized trans people who have histories of carding and other legitimized forms of distrust toward providing personal information to the government [66].

Considering that many forms of identification such as drivers licenses were created to identify and exclude Black and Indigenous people, creating more surveillance categories risks more mistrust in government systems for communities who have faced oppression and

marginalization [66–68]. These are all factors to consider in our recommendation to review the current gendering systems and make changes to ensure accessible and safe services for trans and non-binary service users who seek STI testing.

Finally, TGBM communities would benefit from a publicly available document for the trans community to access that outlines the steps needed currently to change their assigned gender within internal systems in healthcare institutions such as clinics and hospitals. As stated by the OWJN, the ability for clients to change internal documents exists in Ontario, however there are no resources outlining these steps for trans people to know that this change would be an option available to the general public. These documents would be paired with educational materials for administrators who may not be aware that clients are able to change their gender internally, or do not know the process of this change. We also recommend the initial step of educational documents to navigate and inform service users and service providers to create a more inclusive healthcare environment that reduces barriers to access healthcare for the trans community.

## Strengths and limitations

This paper's strength is that it focuses solely on TGBM's testing barriers, where most work in examining barriers to testing has included all transgender identities, or only trans women and HIV testing [6, 69]. The use of Peer and Community-Based Participatory Research is also an asset to this paper to support the participants and conduct culturally competent research.

This study had several limitations that should be noted. The barriers identified by the TGBM we interviewed may not be representative of the larger community, given the small purposive sample, and that participants had to have been tested for an STI in the twelve months prior to their involvement in the study. The larger study was to ascertain the accept-ability of online testing in Ontario compared to in-clinic testing and therefore our participants needed to be tested recently to provide this comparison. However, this inclusion criteria meant that TGBM who participated in the study overcame the barriers they faced, at least in part, to be tested. As such, we were not able to interview TGBM who tried to be tested in the last 12 months but were not able to due to the barriers they faced. We believe that this prelimi-nary data remains important and notable for future research to draw upon to study the barri-ers TGBM have faced when they sought testing.

Another limitation of the study concerns the relative homogeneity of the sample, with almost all participants residing in large urban centers in Ontario, lack of trans elders, and par-ticipants having an overall high level of education. It is also important to note that participants would have had to 'out' themselves as trans to cisgender GBM during the course of their inter-view or focus group to relay experiences that identified them as trans. Self-disclosure to their cisgender (or perceived cisgender) peers may have created barriers to disclosure of their iden-tity for TGBM in the conversation, or resulted in TGBM identified participants not speaking to how their TGBM identify effected their testing habits. Additionally, because of COVID-19 the focus groups and interviews were conducted over Microsoft Teams, excluding those with poor or no internet access, or limited privacy to engage in the study from home.

These limitations were considered throughout the research process, and steps were taken to mitigate these barriers. Peer researchers who identified as TGBM were included in focus groups and interviews and used methods of self-disclosure to open space for TGBM partici-pants to share their own identities and experiences. To offset the sample of majority white par-ticipants in the first round of recruitment, a second round was conducted specifically to include Black and Indigenous identities and their unique experiences.

## Conclusion

Within this sample of TGBM in Ontario, Canada we found that there were trans-specific barriers to STI testing for TGBM involving lack of inclusive clinic environments, lack of trans competent providers, and incongruent documentation. We put forth three primary recommendations to reduce these barriers which include: clinics providing all-gender clinic hours, increased LGBTQ+ clinics and provider training in transgender health, and creating a review for a publicly-funded health system that is that provides accurate and inclusive options for trans communities.

## Supporting information

**S1 Fig. Focus group guide for service users of STI testing services.**
(TIF)

## Acknowledgments

The authors would like to thank Joshun Dulai, Ryan Lisk, Mark Gaspar, and Hsiu-Ju Chang for their assistance in the process of this research and development of this manuscript. We would also like to thank the GetCheckedOnline community advisory board for their help in the research process and consultation on the content of this manuscript.

## Author Contributions

**Conceptualization:** Praney Anand, Oralia Gómez-Ramirez, Kinnon R. MacKinnon, Catherine Worthington, Mark Gilbert, Daniel Grace.

**Data curation:** Mackenzie Stewart, Heeho Ryu, Ezra Blaque, Praney Anand, Oralia Gómez-Ramirez, Kinnon R. MacKinnon, Daniel Grace.

**Formal analysis:** Mackenzie Stewart, Heeho Ryu, Ezra Blaque.

**Funding acquisition:** Catherine Worthington, Mark Gilbert, Daniel Grace.

**Investigation:** Mackenzie Stewart, Heeho Ryu, Ezra Blaque, Abdi Hassan, Catherine Worthington, Daniel Grace.

**Methodology:** Mackenzie Stewart, Heeho Ryu, Ezra Blaque, Abdi Hassan.

**Project administration:** Mackenzie Stewart, Praney Anand.

**Resources:** Mackenzie Stewart, Praney Anand.

**Software:** Mackenzie Stewart.

**Supervision:** Mackenzie Stewart, Praney Anand, Daniel Grace.

**Validation:** Mackenzie Stewart, Praney Anand.

**Visualization:** Mackenzie Stewart, Praney Anand.

**Writing – original draft:** Mackenzie Stewart.

**Writing – review & editing:** Mackenzie Stewart, Heeho Ryu, Ezra Blaque, Abdi Hassan, Praney Anand, Oralia Gómez-Ramirez, Kinnon R. MacKinnon, Catherine Worthington, Mark Gilbert, Daniel Grace.

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
