## [Decision Letter · Decision Letter 0]

29 Jul 2022

PONE-D-22-17296Cisnormativity as a structural barrier to STI testing for trans masculine, two-spirit, and non-binary people who are gay, bisexual, or have sex with menPLOS ONE

Dear Dr. Stewart,

Thank you for submitting your manuscript to PLOS ONE. After careful consideration, we feel that it has merit but does not fully meet PLOS ONE’s publication criteria as it currently stands. Therefore, we invite you to submit a revised version of the manuscript that addresses the points raised during the review process.

 In your revised submission, please respond to each of the comments from each reviewer. We look forward to receiving your revision.

We look forward to receiving your revised manuscript.

Kind regards,

Amy Michelle DeBaets, PhD

Academic Editor

PLOS ONE

Journal Requirements:

Reviewers' comments:

Reviewer's Responses to Questions

**Comments to the Author**

1. Is the manuscript technically sound, and do the data support the conclusions?

Reviewer #1: Yes

Reviewer #2: Yes

2. Has the statistical analysis been performed appropriately and rigorously? 

Reviewer #1: N/A

Reviewer #2: N/A

3. Have the authors made all data underlying the findings in their manuscript fully available?

Reviewer #1: No

Reviewer #2: No

4. Is the manuscript presented in an intelligible fashion and written in standard English?

Reviewer #1: No

Reviewer #2: Yes

5. Review Comments to the Author

Reviewer #1: 1) The GCO first comes up in the method. It seems that this might have been a key focus or driver of the study? If so, this should be signalled earlier in the paper and more details given about the GCO

2) It is noted in the method that cis and trans participants were looked at comparatively. It could be useful to briefly note overlaps between the two groups, to further highlight what was unique to the trans participants.

3) It reads as though each theme has sub themes, but they are not highlighted as such. I wondered if it might be helpful to the reader to use sub-sub headings?

4) It might be helpful to include a note about Ian's reference to heteronormative clinics for the lay reader? What Ian seems to be describing is cisgenderism, not heteronormativity, so the reference by him to heteronormative clinics may be confusing for some.

5) Is it a 'trans identity' or is it 'being trans'? Same, is it 'gender identity' or just 'gender' (it is rare to refer to cis people's 'gender identity', rather just 'gender'). If you mean gender modality (as per Florence Ashley, then say that)

6) The reference to 'Sean stated in a separate interview' makes the method a little unclear. Did some participants attend both a focus group and an interview?

7) Some of the block quotes are a little long and could be edited somewhat

8) I was left wondering if the trans participants had anything unique to say about the GCO?

9) The language of 'match' is fraught (and Ansara defines it as cisgenderist). Would it be clearer to say 'documentation did not reflect their gender' throughout or similar?

10) Some of the findings (particularly the need to educate/lack of education on the part of health providers) are very similar to other research on trans people's experiences with HCPs. It would be worth noting this similarity as a common theme beyond the narrow focus on STI testing.

Reviewer #2: General comment:

This is an important study that offers practical recommendations to improve the experience of STI testing in the TGBM population. The methodology is sound, clearly described and appears to have been conducted with cultural humility and reflexivity. The authors’ implementation of CBPR as the chosen research framework, and their clear focus on research ethics principles is to be highly commended. Most importantly, the authors have offered an analysis of the unique perspectives of a minority group that is underrepresented in the literature, so the work has great value on this account alone.

Most of my recommendations are minor, however one is of greater significance.

Major issue

The major premise of the study, as stated in the first line of the abstract (line 25) and elsewhere throughout the paper, is that TGBM are under-tested for sexually transmitted infections. The logical argument of the paper as presented in the abstract is that:

(1) TGBM are under-tested for STI

(2) this is likely due to (complex and intersecting) barriers to STI testing

(3) this is likely due to TGBM-specific barriers to STI testing, and

(4) if these barriers are identified we can make changes to facilitate uptake of STI testing in TGBM

And while I found the results and discussion sections to be compelling and thorough in their answering of premises 2 – 4, adequate evidence was not provided for premise 1: that testing rates in this community are low. I recommend that the introduction undergo major revision to meet this point and to build to the study rationale. More broadly, the introduction would benefit from covering the literature in more detail to appropriately contextualise this study.

Following from this, while readers are adequately informed (in the introduction) of the risk-levels of TGBM contracting STI when compared to cisgender men, the importance of this in relation to the aims of the study must be clarified: that sexual health care (and removing barriers to access) is important for TGBM populations as they bear a disproportionate global burden of STI etc.

If there is not enough evidence to support the assumption that TGBM are under-tested for STI then this must be appropriately addressed within the study and the basis for its objectives will likely need to be reframed.

Some articles that may be useful include:

Sharma A, Kahle E, Todd K, Peitzmeier S, Stephenson R. Variations in Testing for HIV and Other Sexually Transmitted Infections Across Gender Identity Among Transgender Youth. Transgend Health. 2019 Feb 20;4(1):46-57. doi: 10.1089/trgh.2018.0047. PMID: 30805557; PMCID: PMC6386078.

Reisner SL, Poteat T, Keatley J, Cabral M, Mothopeng T, Dunham E, Holland CE, Max R, Baral SD. Global health burden and needs of transgender populations: a review. Lancet. 2016 Jul 23;388(10042):412-436.

doi: 10.1016/S0140-6736(16)00684-X. Epub 2016 Jun 17. PMID: 27323919; PMCID: PMC7035595.

Rosenberg S, Callander D, Holt M, Duck-Chong L, Pony M, Cornelisse V, et al. (2021) Cisgenderism and transphobia in sexual health care and associations with testing for HIV and other sexually transmitted infections: Findings from the Australian Trans & Gender Diverse Sexual Health Survey. PLoS ONE 16(7): e0253589. https://doi.org/10.1371/journal.pone.0253589

World Health Organization. Regional Office for the Western Pacific. (‎2013)‎. Regional assessment of HIV, STI and other health needs of transgender people in Asia and the Pacific. WHO Regional Office for the Western Pacific. https://apps.who.int/iris/handle/10665/207686

Minor Issues

In the results section cisnormativity and heteronormativity are identified as overarching barriers encompassing the identified themes. However, in the abstract section only cisnormativity is mentioned.

Lines 38 – 39: Consider “among [this] TGBM population…”

Clarification here may be helpful to the reader - have these themes been identified from the authors’ research generally (as applying to the TGBM population in a broader sense) or have these themes been identified from their respondent data. If it is the latter, I suggest that this is specified.

Lines 41 – 42: “Institutional barriers to testing appear to be factors shaping the historical under-testing for STI in the TGBM population due to their inherent cisnormativity.” This sentence doesn’t necessarily follow, consider revising.

Consider moving the ‘however clause’ in line 55: “[however] TGBM may require specialized care and different tests in comparison to cisgender GBM…” to the end of line 50-51. As a stand-alone comment it doesn’t frame the following points I think the authors are trying to make e.g. that GBM have higher risk profiles for STI than the general population (assumed, not stated), and that because TGBM can require specialised tests this may prove a barrier to testing. The reader is working hard here to gather implied meaning.

Lines 72 – 85: it may be useful to compare this data to general or total population to contextualise the percentage figures.

Line 73 does not seem to follow from Line 72 despite the “also” – clarification as to the link between these statements would be useful for readers.

Line 98: “healthcare [to] trans patients”

Line 101: consider “fear of [a] positive result” or result[s] plural

Lines 103 – 105: tense appears to have been swapped from present to past tense

Lines 107 – 121: The authors present an important point that non-HIV STI are overlooked in the literature around TGBM STI testing. The example of some non-STI testing methods presenting a likely barrier due to low acceptability of genital swabs should be evidenced, despite seeming intuitive. Furthermore, this point becomes repeated, and the paragraph becomes clunky. The authors could consider condensing or removing some lines here e.g. line 117-119.

Line 547: the end of the sentence appears to have been cut off.

Lines 618 – 622: This is an important point re. excluding participants who haven’t been tested for an STI in the past 12 months. I can understand why it was chosen as an exclusion criterion in the context of the larger study but given its impact as a limitation on this work further explanation of this is warranted here.

Conclusion:

I enjoyed reading this study and think that it will make an important contribution to transgender persons' experiences of STI testing. I suggest major revision of the introduction section and minor revisions elsewhere.

6. PLOS authors have the option to publish the peer review history of their article (what does this mean?). If published, this will include your full peer review and any attached files.

Reviewer #1: No

Reviewer #2: No

---

## [Author Response · Author response to Decision Letter 0]

14 Sep 2022

We would like to thank the reviewers for their supportive and helpful comments on our manuscript “Cisnormativity is a structural barrier to STI testing for transmasculine, two-spirit and non-binary people who are gay, bisexual or have sex with men.” We are thankful for the opportunity to revise this manuscript and are submitting an updated version of this paper as requested. A detailed response to reviewers’ comments can be found below.

Editor Comments and Additional Requirements

The style guidelines have been closely reviewed and required edits have been made throughout to ensure we meet style requirements. 

Thank you for this note, we will ensure that these two entries are congruent upon resubmission.

Thank you for this comment. We have reviewed the policies in detail, including guidance for qualitative research articles with highly sensitive data, including clinical data. These are highly sensitive interviews conducted with marginalized groups with potentially identifying information which means that full transcripts cannot be made available for privacy reasons. Doing so would also violate our REB commitments. In line with the examples for acceptable text online, including recent publications in this journal, we have included the following sentences to describe data availability: 

“All relevant data are within the manuscript and its Supporting Information files. A detailed interview guide, which was used for individual interviews and focus groups, is provided.” 

Reviewer #1 

1. The GCO first comes up in the method. It seems that this might have been a key focus or driver of the study? If so, this should be signalled earlier in the paper and more details given about the GCO. 

Thank you for this feedback. A descriptive paragraph of the project’s aim and the GetCheckedOnline (GCO) program have been added to the methods section which now reads: 

“Our TGBM data are part of a larger study examining cis and trans GBM’s STI testing experiences in Ontario. The objective of the larger study was to gauge the acceptability and population-specific benefits of an online testing program. We used the online testing program GetCheckedOnline (GCO), currently available in parts of British Columbia, Canada, as a model for what online testing could look like in Ontario. GCO is currently operated by the British Columbia Centre for Disease Control (BCCDC). The process of testing with GCO is as follows: those seeking testing create an account on getcheckedonline.com; fill out a sexual behavior-based questionnaire; have tests recommended to them based on their behavior; they are sent an electronic lab requisition that they can then bring into select locations of Lifelabs, a Canadian-based lab corporation; they present the lab requisition on their phone or by paper, they have their specimens collected and are notified through the online system if their results are negative. If results are positive or inconclusive, they are given the number of a public health nurse to contact who will connect them with further care and resources.”

2. It is noted in the method that cis and trans participants were looked at comparatively. It could be useful to briefly note overlaps between the two groups, to further highlight what was unique to the trans participants.

We appreciate the note to add more clarity between participant groups in our study to further highlight the circumstances unique to TGBM. We added a section describing the overlapping barriers: “These barriers were specific to TGBM, and were compounded by the barriers we found in both cis and trans GBM, which included: inaccessible or inconvenient clinic hours, lack of anonymity in testing environment, fear of stigma from providers about sexual behaviors or practices, as well as low levels of trust in providers to be informed about GBM identities.”

3. It reads as though each theme has sub themes, but they are not highlighted as such. I wondered if it might be helpful to the reader to use sub-sub headings?

Thank you for this suggestion. Under “cisnormative testing environments” we added the following subheadings: gender segregated clinics, trans-inclusive terminology, institutional cis and heteronormativity. “Lack of trans competency” now includes the subheadings: provider education, assumptions from providers, and lack of TGBM knowledge. No subheadings were added under documentation barriers as this was one larger theme with no subthemes. 

4. It might be helpful to include a note about Ian's reference to heteronormative clinics for the lay reader? What Ian seems to be describing is cisgenderism, not heteronormativity, so the reference by him to heteronormative clinics may be confusing for some.

We have made the suggested revisions to add clarity for the reader. This reference has been changed to: 

“For example, Ian noted his experience with what he termed “normative” clinics, and while he specifically names heteronormatvity in these interactions; his experiences also reflect cisnormativity and how they impacted his desire to be tested.” 

5. Is it a 'trans identity' or is it 'being trans'? Same, is it 'gender identity' or just 'gender' (it is rare to refer to cis people's 'gender identity', rather just 'gender'). If you mean gender modality (as per Florence Ashley, then say that)

Thank you for this suggestion. We have adapted the language of gender modality and made the change throughout the document to replace the use of ‘gender identity’.

6. The reference to 'Sean stated in a separate interview' makes the method a little unclear. Did some participants attend both a focus group and an interview?

For clarity this has been changed the language to: “Sean shared similar sentiments to Ian in their own interview, and stated that they had also felt it was personally difficult to be tested”.

7. Some of the block quotes are a little long and could be edited somewhat

Thank you for this feedback. Some block quotes have been edited for clarity and brevity. The new revised quotes read: 

“I asked a doctor who gave me results for something and they had no idea about the impact [of an] STI issue that’s very specific to our community as transmen. And had it been the doctor or the specialist who was a gay man that works at the HIV clinic he would’ve been ‘oh yeah, like you know, give [the test]’, right?” 

“Definitely in terms of different types of testing like I have gotten an HIV rapid test because it’s very like comfortable in how quick and easy it is but personally as a trans person I wouldn’t go to just any doctor for every STI test. I feel like they would need to be informed. […] I need to feel comfortable with the whole situation and I can’t just be thrown into it.”

“Darren: Well the only thing that I want to share are my tests are not like forced because I can choose to have it, but they are sent in all the time from my doctor [it’s] part of the – her service, like she’s always taking care of them. […] 

Damian: Yeah, I can jump in. I think sort of similar to what Darren said that my doctor’s always like “You need to get tested.” It’s not like they’re forcing it, but they’re always reminding me, and I think that comes from having a very like LGBT [competent] family physician.”

8. I was left wondering if the trans participants had anything unique to say about the GCO?

We thank their reviewer for their curiosity about the overall project. The acceptability of GCO best fits in a separate manuscript submission to reduce the length of this one, and have this manuscript focus solely on the barriers to in-person testing models. 

9. The language of 'match' is fraught (and Ansara defines it as cisgenderist). Would it be clearer to say 'documentation did not reflect their gender' throughout or similar?

Thank you for this feedback, we have removed the language of “matching” documentation. Additionally, we added the following note: “We acknowledge that gender markers on these documents are static texts that do not reflect the fluidity of identity, and cannot accurately reflect the gender modality of people holding these documents; more can be done to support trans people in reflecting their ideal gender representation on these documents. However, reconciling the needs of trans people and needs of public health surveillance need to both be accounted for when making any changes to the current systems.”

10. Some of the findings (particularly the need to educate/lack of education on the part of health providers) are very similar to other research on trans people's experiences with HCPs. It would be worth noting this similarity as a common theme beyond the narrow focus on STI testing.

We appreciate this recommendation. We have inserted previous studies describing similar findings. The paragraph now reads: 

“Previous studies have indicated that providers lack education on trans patients, not exclusive to TGBM, across healthcare. 57-59 As found in our study, having uneducated providers made trans people less likely to want to access care.59 One study found that 24% of their sample of trans patients had to educate their healthcare providers about trans people and health over the course of their visit.52 However, the other finding from this study indicated that this need to educate directly impacted trans patient’s mental health. 59 Patients who had to educate their providers were more likely to experience depression and anxiety. 59 These findings are not exclusive to North America and have global implications; the World Health Organization conducted an assessment of transgender people in Asia and the Pacific and found that trans people who had access to healthcare providers were stigmatized and faced discrimination from their providers due to lack of trans knowledge.53 Another study in Vietnam found that providers had knowledge gaps and were unprepared to treat transgender clients. 60 In general, healthcare for trans people is falling short of adequate and informed care, and sexual and reproductive healthcare follows this trend.”

Reviewer #2: 

11. General comment: This is an important study that offers practical recommendations to improve the experience of STI testing in the TGBM population. The methodology is sound, clearly described and appears to have been conducted with cultural humility and reflexivity. The authors’ implementation of CBPR as the chosen research framework, and their clear focus on research ethics principles is to be highly commended. Most importantly, the authors have offered an analysis of the unique perspectives of a minority group that is underrepresented in the literature, so the work has great value on this account alone.

We thank the reviewer for their supportive comments and pleased they see the value of this contribution to the literature and in our methodology. No revisions are required based on this comment.

12. Most of my recommendations are minor, however one is of greater significance. Major issue: The major premise of the study, as stated in the first line of the abstract (line 25) and elsewhere throughout the paper, is that TGBM are under-tested for sexually transmitted infections. The logical argument of the paper as presented in the abstract is that: (1) TGBM are under-tested for STI; (2) this is likely due to (complex and intersecting) barriers to STI testing; (3) this is likely due to TGBM-specific barriers to STI testing, and; (4) if these barriers are identified we can make changes to facilitate uptake of STI testing in TGBM. 

And while I found the results and discussion sections to be compelling and thorough in their answering of premises 2 – 4, adequate evidence was not provided for premise 1: that testing rates in this community are low. I recommend that the introduction undergo major revision to meet this point and to build to the study rationale. More broadly, the introduction would benefit from covering the literature in more detail to appropriately contextualise this study.

Thank you for the opportunity to clarify and add nuance to premise 1. We have addressed this point in the introduction. The introduction now reads: 

“The limited available data currently indicates that TGBM are under-tested for STI while having a similar risk profile as their cisgender MSM counterparts. 6, 7 Despite having high levels of condomless sex, 43% of TGBM in Ontario had never been tested for HIV, leading to possible underdiagnosis of HIV in the TGBM population in Ontario.8 In the general Candian GBM population, approximately 17% had never been tested, a dramatic difference from their TGBM peers in Ontario.9 

The trends of transgender men, including those who have sex with only women, being under-tested also have been tracked worldwide. In Thailand a sexual health center that services only transgender clients, only 5% of service users were trans masculine people who primarily accessed the clinic for gender-affirming hormones, not for STI testing. 10 Most studies that analyze transgender sexual health focus on trans women or people assigned male at birth.11 This lack of data includes the CDC citing only trans women’s risk, prevalence and prevention on their website and reports. 12 This dearth of data indicates that the TGBM community currently lacks uptake of available services and needs targeted interventions to increase access to STI testing.11 The impact of targeted programming would be to diagnose more STI in TGBM and provide accurate rates of STI in this community.”

13. Following from this, while readers are adequately informed (in the introduction) of the risk-levels of TGBM contracting STI when compared to cisgender men, the importance of this in relation to the aims of the study must be clarified: that sexual health care (and removing barriers to access) is important for TGBM populations as they bear a disproportionate global burden of STI etc. If there is not enough evidence to support the assumption that TGBM are under-tested for STI then this must be appropriately addressed within the study and the basis for its objectives will likely need to be reframed.

Some articles that may be useful include:

Sharma A, Kahle E, Todd K, Peitzmeier S, Stephenson R. Variations in Testing for HIV and Other Sexually Transmitted Infections Across Gender Identity Among Transgender Youth. Transgend Health. 2019 Feb 20;4(1):46-57. doi: 10.1089/trgh.2018.0047. PMID: 30805557; PMCID: PMC6386078.

Reisner SL, Poteat T, Keatley J, Cabral M, Mothopeng T, Dunham E, Holland CE, Max R, Baral SD. Global health burden and needs of transgender populations: a review. Lancet. 2016 Jul 23;388(10042):412-436.

doi: 10.1016/S0140-6736(16)00684-X. Epub 2016 Jun 17. PMID: 27323919; PMCID: PMC7035595.

Rosenberg S, Callander D, Holt M, Duck-Chong L, Pony M, Cornelisse V, et al. (2021) Cisgenderism and transphobia in sexual health care and associations with testing for HIV and other sexually transmitted infections: Findings from the Australian Trans & Gender Diverse Sexual Health Survey. PLoS ONE 16(7): e0253589. https://doi.org/10.1371/journal.pone.0253589

World Health Organization. Regional Office for the Western Pacific. (‎2013)‎. Regional assessment of HIV, STI and other health needs of transgender people in Asia and the Pacific. WHO Regional Office for the Western Pacific. https://apps.who.int/iris/handle/10665/207686

Thank you for this incredibly helpful feedback and references. Following the previous addition, we added comparative statistics for transgender men to help frame their lack of testing to their cisgender peers. We have also added your suggested references throughout the manuscript. The additional paragraph now reads: 

“There is a paucity of data surrounding STI research for TGBM, including the barriers preventing this population from accessing testing.19, 20 Often, studies on TGBM infection risk are focused on HIV, and the literature suggests an estimation of HIV prevalence in all trans men is between 0% to 3%, with 0.6% of Ontarian trans men reporting an HIV infection.,21-23 Previous studies in Ontario noted that 45% of trans men engaged in condomless anal or vaginal sex with a partner of unknown serostatus, which is considered a high-risk activity for contracting HIV.21, 24 In 2018 the United States Centers for Disease Control (CDC) identified that .5% of 2,364 trans men in their sample were diagnosed with HIV, which is more than double the rate of cisgender women at .2% out of 4,753,672 , and less than cisgender men at .9% out of 4,534,426, and far less than trans women who had the highest rate of HIV diagnosis at 2.7% of 13,154.25 We are using the 2018 data because there have been no updated statistics on the CDC website since 2018 about diagnosis rates in trans men. The 2018 statistics may be inaccurate due to the noted lack of testing in this population and the fact that in these categories in the number of trans men in their sample was exceedingly lower than the other populations being sampled. Additionally, these numbers are inclusive of HIV only and no other STI, which does not give a broad or accurate look at trans men’s risk levels of STI acquisition.”

14. Minor Issues: In the results section cisnormativity and heteronormativity are identified as overarching barriers encompassing the identified themes. However, in the abstract section only cisnormativity is mentioned.

We appreciate this observation. The abstract now reads: “When questioned about past experiences with testing, TGBM participants reported several barriers to STI testing within current testing models in Ontario due to cisnormativity and heteronormativity. Cisnormativity is the assumption that everyone identifies as the gender they were assigned at birth, and those who do not are considered “abnormal”, while heteronormativity is when it is assumed that everyone is heterosexual.”

15. Lines 38 – 39: Consider “among [this] TGBM population…”

Clarification here may be helpful to the reader - have these themes been identified from the authors’ research generally (as applying to the TGBM population in a broader sense) or have these themes been identified from their respondent data. If it is the latter, I suggest that this is specified.

We appreciate the note for clarity. We have changed the language to “From our research we identified three overarching themes concerning testing barriers among TGBM participants” to reflect that these themes were identified from our respondent data. 

16. Lines 41 – 42: “Institutional barriers to testing appear to be factors shaping the historical under-testing for STI in the TGBM population due to their inherent cisnormativity.” This sentence doesn’t necessarily follow, consider revising.

We have revised this to “Inherent cis and heteronormativity in healthcare institutions appear to be factors shaping the historical under-testing for STI in the TGBM population”

17. Consider moving the ‘however clause’ in line 55: “[however] TGBM may require specialized care and different tests in comparison to cisgender GBM…” to the end of line 50-51. As a stand-alone comment it doesn’t frame the following points I think the authors are trying to make e.g. that GBM have higher risk profiles for STI than the general population (assumed, not stated), and that because TGBM can require specialised tests this may prove a barrier to testing. The reader is working hard here to gather implied meaning.

Thank you for this note about clarity. We have moved line 55 to after 50-51 so the paragraph now reads: “Despite these variations in terminology and identity, many TGBM have similar risk profiles for sexually transmitted infections (STI) as cisgender gay, bisexual, and other men who have sex with men (GBM).2 However, TGBM may require specialized care and different tests in comparison to cisgender GBM, depending on their body type and what kind of sex they are having.”

18. Lines 72 – 85: it may be useful to compare this data to general or total population to contextualise the percentage figures.

We have added a comparison for context of TGBM being under-tested. This now reads: “The limited available data currently indicates that TGBM are under-tested for STI while having a similar risk profile as their cisgender MSM counterparts. 6, 7 Despite having high levels of condomless sex, approximately 43% of TGBM in Ontario had never been tested for HIV, leading to possible underdiagnosis of HIV in the TGBM population in Ontario.8 In the general Canadian GBM population, approximately 17% had never been tested, a dramatic difference from their TGBM peers in Ontario.9”

19. Line 73 does not seem to follow from Line 72 despite the “also” – clarification as to the link between these statements would be useful for readers.

We appreciate this note and have revised the paragraph to address a previous reviewer comment. This section now reads: 

“There is a paucity of data surrounding STI research for TGBM, including the barriers preventing this population from accessing testing.19, 20 Often, studies on TGBM infection risk are focused on HIV, and the literature suggests an estimation of HIV prevalence in all trans men is between 0% to 3%, with 0.6% of Ontarian trans men reporting an HIV infection.,21-23 Previous studies in Ontario noted that 45% of trans men engaged in condomless anal or vaginal sex with a partner of unknown serostatus, which is considered a high-risk activity for contracting HIV.21, 24 In 2018 the United States Centers for Disease Control (CDC) identified that .5% of 2,364 trans men in their sample were diagnosed with HIV, which is more than double the rate of cisgender women at .2% out of 4,753,672 , and less than cisgender men at .9% out of 4,534,426, and far less than trans women who had the highest rate of HIV diagnosis at 2.7% of 13,154.25 We are using the 2018 data because there have been no updated statistics on the CDC website since 2018 about diagnosis rates in trans men. The 2018 statistics may be inaccurate due to the noted lack of testing in this population and the fact that in these categories in the number of trans men in their sample was exceedingly lower than the other populations being sampled. Additionally, these numbers are inclusive of HIV only and no other STI, which does not give a broad or accurate look at trans men’s risk levels of STI acquisition. 

The data related to TGBM and STI that are not HIV primarily come from convenience samples. Participants in a study in New England also noted that they considered themselves to be “moderately high risk” for acquiring an STI.26”

20. Line 98: “healthcare [to] trans patients”

Thank you for catching this typo; we have revised the text accordingly. 

21. Line 101: consider “fear of [a] positive result” or result[s] plural

We have revised this statement to “fear of positive results”.

22. Lines 103 – 105: tense appears to have been swapped from present to past tense

Thank you for noting the tense change, we have revised the paragraph to read: 

“STI testing presents its own unique set of barriers for TGBM including: the fear of positive results delaying transition, mistrust of sexual health providers based on previous negative experiences, perceived low risk of trans patients acquiring STI by providers due to the lack of knowledge of TGBM sex practices, assumption of clients being cisgender leading to lack of adequate care, and providers not having trans-specific sexual health knowledge which impedes testing or leads to lack of appropriate testing for TGBM. 29, 35”

23. Lines 107 – 121: The authors present an important point that non-HIV STI are overlooked in the literature around TGBM STI testing. The example of some non-STI testing methods presenting a likely barrier due to low acceptability of genital swabs should be evidenced, despite seeming intuitive. Furthermore, this point becomes repeated, and the paragraph becomes clunky. The authors could consider condensing or removing some lines here e.g. line 117-119.

We want to thank the reviewer for this note, as it has highlighted to us the lack of research surrounding the acceptability of different methods of STI testing. We have streamlined this paragraph and cited the limited research in the field that supports the arguments we are advancing here. The streamlined paragraph now reads: 

“We were unable to identify any current research that indicated the difference in acceptability between STI tests with genital swabs, and those that do not require genital swabs for TGBM. However, it is the anecdotal experience of the authors who identify as TGBM, as well as an inductive finding through our research that these differences in testing methods may affect testing frequency and increase testing avoidance amongst TGBM. Additionally, previous studies examining trans people’s experiences with reproductive healthcare indicated that internal examinations or insertions may present a barrier for trans people accessing health resources due to their perceived incongruence with a patient’s gender identity. One study found that birth control methods that require insertion, such as an IUDs or vaginal rings, may cause or aggravate dysphoria for some trans people. 38 Another study examining the experiences of trans people with cervixes accessing pap tests reported that a major barrier to trans participants who sought testing was reconciling their masculine identity and what they saw as a ‘feminizing’ or ‘feminine’ procedure, and those who were on the far end of masculinity struggled with the gender dysphoria they faced from a pap test.39 Based on these studies, we conclude that similar barriers may be faced due to internal STI swabs in the TGBM community and believe there should be further research and action to reduce these the distress internal exams may cause for TGBM patients.”

24. Line 547: the end of the sentence appears to have been cut off.

Thank you for catching this error. The complete sentence now reads: “Our findings as well as other literature demonstrate that for TGBM, living at the intersection of being both a gender and sexual minority has created unique barriers for those who seek care from any provider, including sexual and reproductive health providers.”

25. Lines 618 – 622: This is an important point re. excluding participants who haven’t been tested for an STI in the past 12 months. I can understand why it was chosen as an exclusion criterion in the context of the larger study but given its impact as a limitation on this work further explanation of this is warranted here.

Two explanations have been added and this paragraph now reads: 

“The larger study was to ascertain the acceptability of online testing in Ontario compared to in-clinic testing and therefore our participants needed to be tested recently to provide this comparison. However, this inclusion criteria meant that TGBM who participated in the study overcame the barriers they faced, at least in part, to be tested. As such, we were not able to interview TGBM who tried to be tested in the last 12 months but were not able to due to the barriers they faced. We believe that this preliminary data remains important and notable for future research to draw upon to study the barriers TGBM have faced when they sought testing.”

26. Conclusion: I enjoyed reading this study and think that it will make an important contribution to transgender persons' experiences of STI testing. I suggest major revision of the introduction section and minor revisions elsewhere.

We are grateful for the supportive feedback from the reviewer and believe that the revisions above that we made from the feedback have helped strengthen and add clarity to our manuscript. 

Thank you again for the opportunity to revise our submission and add clarity to the prose and points made in the manuscript. We look forward to hearing from you.

---

## [Decision Letter · Decision Letter 1]

25 Oct 2022

Cisnormativity as a structural barrier to STI testing for trans masculine, two-spirit, and non-binary people who are gay, bisexual, or have sex with men

PONE-D-22-17296R1

Dear Dr. Stewart,

We’re pleased to inform you that your manuscript has been judged scientifically suitable for publication and will be formally accepted for publication once it meets all outstanding technical requirements.

Kind regards,

Amy Michelle DeBaets, PhD

Academic Editor

PLOS ONE

Additional Editor Comments (optional):

Reviewers' comments:

Reviewer's Responses to Questions

**Comments to the Author**

1. If the authors have adequately addressed your comments raised in a previous round of review and you feel that this manuscript is now acceptable for publication, you may indicate that here to bypass the “Comments to the Author” section, enter your conflict of interest statement in the “Confidential to Editor” section, and submit your "Accept" recommendation.

Reviewer #1: All comments have been addressed

Reviewer #2: All comments have been addressed

2. Is the manuscript technically sound, and do the data support the conclusions?

Reviewer #1: Yes

Reviewer #2: Yes

3. Has the statistical analysis been performed appropriately and rigorously? 

Reviewer #1: N/A

Reviewer #2: N/A

4. Have the authors made all data underlying the findings in their manuscript fully available?

Reviewer #1: Yes

Reviewer #2: No

5. Is the manuscript presented in an intelligible fashion and written in standard English?

Reviewer #1: Yes

Reviewer #2: Yes

6. Review Comments to the Author

Reviewer #1: (No Response)

Reviewer #2: The authors have addressed all comments in detail. In particular the changes made to the introduction in order to clarify and contextualise the under-testing of STI among TGBM has greatly strengthened the manuscript as a whole. I enjoyed reading the revised manuscript - my congratulations to the authors!

7. PLOS authors have the option to publish the peer review history of their article (what does this mean?). If published, this will include your full peer review and any attached files.

Reviewer #1: No

Reviewer #2: No

---

## [Editor Report · Acceptance letter]

3 Nov 2022

PONE-D-22-17296R1 

Cisnormativity as a structural barrier to STI testing for trans masculine, two-spirit, and non-binary people who are gay, bisexual, or have sex with men 

Dear Dr. Stewart:

I'm pleased to inform you that your manuscript has been deemed suitable for publication in PLOS ONE. Congratulations! Your manuscript is now with our production department. 

Kind regards, 

on behalf of

Dr. Amy Michelle DeBaets 

Academic Editor

PLOS ONE